# Ability of Delta Radiomics to Predict a Complete Pathological Response in Patients with Loco-Regional Rectal Cancer Addressed to Neoadjuvant Chemo-Radiation and Surgery

**DOI:** 10.3390/cancers14123004

**Published:** 2022-06-18

**Authors:** Valerio Nardone, Alfonso Reginelli, Roberta Grassi, Giovanna Vacca, Giuliana Giacobbe, Antonio Angrisani, Alfredo Clemente, Ginevra Danti, Pierpaolo Correale, Salvatore Francesco Carbone, Luigi Pirtoli, Lorenzo Bianchi, Angelo Vanzulli, Cesare Guida, Roberto Grassi, Salvatore Cappabianca

**Affiliations:** 1Department of Precision Medicine, University of Campania “L. Vanvitelli”, 80138 Naples, Italy; alfonso.reginelli@unicampania.it (A.R.); roberta.grassi@policliniconapoli.it (R.G.); giovanna.vacca@student.unicampania.it (G.V.); giuliana.giacobbe@student.unicampania.it (G.G.); antonio.angrisani@student.unicampania.it (A.A.); alfredo.clemente@student.unicampania.it (A.C.); roberto.grassi@unicampania.it (R.G.); salvatore.cappabianca@unicampania.it (S.C.); 2Italian Society of Medical and Interventional Radiology (SIRM), SIRM Foundation, 20122 Milan, Italy; 3Department of Emergency Radiology, Careggi University Hospital, 50134 Florence, Italy; ginevra.danti12@libero.it; 4Unit of Medical Oncology, Grand Metropolitan Hospital “Bianchi Melacrino Morelli”, 89100 Reggio Calabria, Italy; pierpaolo.correale@ospedalerc.it; 5Unit of Medical Imaging, University Hospital of Siena, 53100 Siena, Italy; salvatore_carbone@ao-siena.toscana.it; 6Sbarro Institute for Cancer Research and Molecular Medicine, Center for Biotechnology, Temple University, Philadelphia, PA 19140, USA; luigipirtoli@gmail.com; 7Niguarda Hospital, University of Milan, 20100 Milan, Italy; lorenzo.bianchi@student.unimi.it (L.B.); angelo.vanzulli@lunimi.it (A.V.); 8Unit of Radiation Oncology, Ospedale del Mare, 80147 Naples, Italy; cesare.guida@aslnapoli1centro.it

**Keywords:** rectal cancer, neoadjuvant chemo-radiation, MRI, texture analysis

## Abstract

**Simple Summary:**

The present study aimed to investigate the possible use of MRI delta texture analysis (D-TA) in order to predict the extent of pathological response in patients with locally advanced rectal cancer addressed to neoadjuvant chemo-radiotherapy (C-RT) followed by surgery. We found that D-TA may really predict the frequency of pCR in this patient setting and, thus, it may be investigated as a potential item to identify candidate patients who may benefit from an aggressive radical surgery.

**Abstract:**

We performed a pilot study to evaluate the use of MRI delta texture analysis (D-TA) as a methodological item able to predict the frequency of complete pathological responses and, consequently, the outcome of patients with locally advanced rectal cancer addressed to neoadjuvant chemoradiotherapy (C-RT) and subsequently, to radical surgery. In particular, we carried out a retrospective analysis including 100 patients with locally advanced rectal adenocarcinoma who received C-RT and then radical surgery in three different oncological institutions between January 2013 and December 2019. Our experimental design was focused on the evaluation of the gross tumor volume (GTV) at baseline and after C-RT by means of MRI, which was contoured on T2, DWI, and ADC sequences. Multiple texture parameters were extracted by using a LifeX Software, while D-TA was calculated as percentage of variations in the two time points. Both univariate and multivariate analysis (logistic regression) were, therefore, carried out in order to correlate the above-mentioned TA parameters with the frequency of pathological responses in the examined patients’ population focusing on the detection of complete pathological response (pCR, with no viable cancer cells: TRG 1) as main statistical endpoint. ROC curves were performed on three different datasets considering that on the 21 patients, only 21% achieved an actual pCR. In our training dataset series, pCR frequency significantly correlated with ADC GLCM-Entropy only, when univariate and binary logistic analysis were performed (AUC for pCR was 0.87). A confirmative binary logistic regression analysis was then repeated in the two remaining validation datasets (AUC for pCR was 0.92 and 0.88, respectively). Overall, these results support the hypothesis that D-TA may have a significant predictive value in detecting the occurrence of pCR in our patient series. If confirmed in prospective and multicenter trials, these results may have a critical role in the selection of patients with locally advanced rectal cancer who may benefit form radical surgery after neoadjuvant chemoradiotherapy.

## 1. Introduction

Preoperative RT, in combination with fluoropyrimidines alone or combination with oxaliplatin, is regarded as the standard of care in the treatment of patients with locally advanced rectal cancer (LARC) before radical surgery (mesorectal excision, TME) [1,2,3,4,5,6]. Chemo-radiotherapy (C-RT), in fact, decreases the rate of local relapse and prolongs the progression-free survival (PFS) of these patients, albeit its effects on overall survival (OS) are still to be proven [7,8]; additionally, it allows for less-invasive surgery with a lower frequency of complications [9]. On the other hand, the evaluation of prognosis indexes, as well as the response assessment of the neoadjuvant chemo-radiation therapy (C-RT) plus total mesorectal excision (TME) [7], still represents a challenge [10], although these topics could be exciting to customize the therapy to the patient needs [11]. Several works in the last few decades suggest the feasibility of a wait-and-see approach in patients with very high surgical risk or trans-anal rectal excision approach if a significant response to CRT is assessed [1,10,12,13]. In this regard, magnetic Resonance Imaging (MRI) seems to be helpful to provide morphological and functional pieces of information that can be used to predict prognosis in pre-treatment patients [14,15,16,17,18,19], but its value in the pre- and post-CRT response assessment is still debated [20,21,22,23,24,25,26,27,28,29]. 

At present, a number of studies highlight the reliability of local staging with pelvic magnetic resonance imaging (MRI) in defining parameters strictly correlated with a high risk of relapse, which include circumferential tumor margins, a possible extramural vascular infiltration, and the presence of metastases to loco-regional nodes [30].

Further, other studies suggest that the additional use of diffusion-weighted imaging (DWI) may also define the tumor tissue cellularity, which, in turn, offers the promising ability to predict the local response to C-RT [31,32,33,34]. 

In more recent times, the development of texture analysis (TA) has allowed researchers to attempt to quantify heterogeneity within the target tumor sites, thus, identifying further valuable parameters, undetectable by naked eye observation [35,36]. TA refers to multiple mathematical models able to provide reliable measurements of heterogeneity within a selected image (texture features). This innovative analysis is currently investigated in several fields and requires a computer quantification of both gray-level intensity and the position of pixels [37,38,39,40,41,42,43,44,45,46,47,48,49]. Several authors are investigating a possible application in the monitoring and research of biomarkers in cancer patients, including those with LARC [27,50,51,52]. In this context, Antunes et al. accurately described, in a retrospective multisite study on radiomic features of rectal cancer for Neoadjuvant C-RT response, how a limited number of four radiomic features extracted from T2w MRI scans allows one to achieve good performance for predicting pCR using Laws and CoLIAgE operators to quantify fluctuations in local image heterogeneity. Those results were achieved independently from MRI scan types (1.5T vs. 3T) [53]. 

The most recent methodological acquisitions have opened the way for the newest TA approaches that take into account the variations in TA parameters recorded at different acquisition times and defined as delta texture analysis or delta radiomics (D-TA) [54,55,56,57,58,59]. It is consequential that these kinds of method may be evaluated as possible items able to investigate the meaning of the reported TA variations in the course of anticancer therapy, including radio/chemo/immuno and target therapy, as well as common neoadjuvant cytoreductive strategies [55,56,57,58,60]. Overall, this promising field of investigation may theoretically lead to a possible oncological treatment, ushering in the future in this field, being more reliable than other imaging approaches in multicenter clinical experimentations [61]. In the last few months, several authors have tested this approach in the context of LARC, with promising results in different endpoints [62,63,64,65,66,67]. In line with these considerations, we designed a retrospective study aiming to evaluate the potential ability of MRI D-TA to predict the frequency of pathological response of patients with LARC undergoing C-RT prior to radical surgery.

## 2. Materials and Methods

### 2.1. Patient Series

This is a retrospective analysis including patients with rectal cancer who received neoadjuvant therapy and then surgery in three different Radiation Oncology Units between January 2013 and December 2019. 

Standard inclusion criteria were taken into consideration for this study (achievable endoscopic/bioptic diagnosis and localized disease at the pre-treatment staging, standard long-course neoadjuvant C-RT protocol, and MR imaging examination at baseline and after the end of C-RT, surgical treatment by TME, and the evaluation of tumor regression grade in the post-surgical report). Patients with MRI examinations without DWI acquisition or lack of post-C-RT MR examination were excluded from the analysis. All the patients gave written consent to anonymous use of their examinations for research scope, and a notification of the study was submitted to the local ethical committee as established by national laws. 

### 2.2. Standard Chemo-Radiation Therapy Protocol

According to the international guidelines, all of the patients received oral chemotherapy with capecitabine (825 mg/m^2^, twice daily for five days/week) which could be delivered daily for five weeks in parallel with the radiotherapy treatment. Radiation therapy volumes focused on the rectum, mesorectum, presacral nodes, and internal iliac nodes [68] with a radiation dosage of 45 Gy, delivered with a conformational radiation technique (3D-CRT) or intensity-modulated RT (IMRT) (5 weekly sessions of 1.8 Gy/daily for five weeks) with further boost dose of 9 Gy (fractioned in five sessions), coned down to tumor with a 2 cm margin and adjacent mesorectal region. Simultaneous integrated boost (SIB-IMRT) was also allowed. 

### 2.3. Magnetic Resonance Imaging

Pelvic MRI examination (1.5-T system, Signa Excite HD, GE Healthcare, Milwaukee, WI, USA for the training dataset, 1.5-T system, Signa Voyager HD, GE Healthcare, Milwaukee, WI, USA and 1.5T system, Achieva XR, Software release 5.3.1, Philips, Amsterdam, The Netherlands, respectively, for the validation datasets) with an eight-channel phased-array coil performed at baseline and 30 ± 15 days after the end of C-RT was available for all the patients included in the study (see Table 1 for the characteristics of the three MRI vendors). 

The imaging protocol was chosen following the European Society of Gastrointestinal Abdominal Radiology (ESGAR) recommendations [5,7,69].

The imaging protocol consisted of high-resolution fast spin-echo (FSE) T2-weighted sequences in the sagittal, axial, and coronal–oblique planes, oriented perpendicularly and parallel to the axial extension of the lesion in the rectal lumen [69]. The DWI is based on the echo-planar spin-echo (SE-EPI) sequence. A fat-saturated pulse was always applied to avoid chemical shift artifacts. Each sequence was acquired in the same axial oblique plane of the T2-weighted images by application of a b-factor and relative apparent diffusion coefficient (ADC) maps.

### 2.4. Surgery and Histopathological Assessment

TME was commonly performed 50 ± 18 days after the end of C-RT protocol. Surgical specimen pathological examinations also reported the tumor regression grade (TRG) established according to the literature (Mandard score: grade 1 complete regression with fibrosis, grade 2 isolated cells in fibrotic tissue, grade 3 cell groups in massive fibrosis, grade 4 high amount of cell groups in fibrotic tissue, grade 5 no regression) [70,71].

### 2.5. Feature Extraction and TA

The gross tumor volume (GTV) was contoured by a radiation oncologist (VN and RG) and confirmed by expert radiologists (SFC, AR) on T2, DWI, ADC sequences. GTV on post C-RT treatment was contoured, taking into consideration the baseline GTV and in selected cases using an elastic fusion approach to assist the contouring. 

The target contouring variations were analyzed by performing two delineations for each patient. The TA parameters were also tested for reliability by using the Intraclass Coefficient Correlation method (ICC). All the analysis were accomplished by using the LifeX Software © (Version 7.2, LITO 22, Deveoped by C. Nioche, INSERM, Paris, France) [72]. TA parameters in particular, included features of gray-level co-occurrence matrix (GLCM), shape parameters, and indices from the gray-level histogram (see Appendix A for the description of the parameters). D-TA was finally calculated as variation ratio of each TA parameter extracted in the two time points, with the formula [(T2-T1)/T1].

### 2.6. Long-Term Follow-Up

All of the patients were scheduled in a post-surgery oncological follow-up program, including a CT-scan and/or MRI repeated every 9–12 weeks at first, and then every six months for the first two years. General examinations with the recording of toxicity, blood cell counts, and chemistry and serum CEA levels were also reported on a three-monthly basis for the first two years. 

### 2.7. Variables’ Selection

A reliability ICC analysis previously identified reliable TA parameters (ICC > 0.70. single measure) including the T2-MRI (7 out of 13 cases; 54%) for DWI-MRI (ten out 13 cases; 77% and ADC-MRI (11 out of 13 cases; 84%) (see Appendix A Materials for the description of the ICC of the texture features).

### 2.8. Endpoints and Statistical Analysis 

The clinical characteristics of the three datasets were tested with Chi-Square analysis (see Table 1). 

### 2.9. Training Dataset

The complete pathological response (pCR, with no viable cancer cells: TRG 1) was chosen as the target statistical endpoint to be statistically correlated with the above-mentioned D-TA parameters. Univariate analysis (univariate logistic regression, with Bonferroni correction for the number of variables) was performed in order to correlate D-TA with the defined endpoint within an internal cohort of patients (see Table 2) observing a correlation larger than 0.80 (Pearson correlation). All of the variables with the inferior univariable correlation with the endpoint were omitted in order to avoid the risk of model overfitting and multicollinearity [73]

The correlation between the significant TA parameters and the endpoint in the multivariate analysis (binary logistic regression) was performed as reported in previous studies from our group [74,75]. Finally, a logistic regression analysis was optimized on a training cohort of patients, and the receiver operating characteristics (ROC) curve was extrapolated from the logistic regression analysis. ROC curve was used to calculate and report the model cut-off.

### 2.10. Dataset Validation

A confirmatory binary logistic regression was repeated in the two validation cohorts, and the corresponding ROC curves were finally extrapolated from the analysis. The cut-off extrapolated by the training dataset was used for the validation dataset. The cut-off of the training model was used to calculate specificity, sensitivity, and accuracy of the model in the training model and validation models. All the statistical analyses were conducted by using SPSS software 23.0 (IBM Corp. Released 2015. IBM SPSS Statistics for Windows, Version 23.0. Armonk, NY: IBM Corp.). and considered as statistically significant when a *p*-value < 0.05 was recorded.

## 3. Results

### 3.1. Patients’ Features

This retrospective analysis was performed on an unmasked sample of 37 consecutive patients in the training dataset and, respectively, 33 and 30 consecutive patients in the two validation datasets. The main characteristics of the three datasets are summarized in Table 2. There was a rate of 27% (10 patients) that showed a pCR in the training dataset and, respectively, 18% (6 patients) and 17% (5 patients) in the two validation datasets. 

### 3.2. Factors Predicting ePD

We performed an analysis of the correlation between the preselected texture analysis parameters, the known prognostic markers, including sex, age, stage, grading, and the pCR, using Bonferroni correction in the training dataset (see Table 3). After Bonferroni correction, the only TA parameters that was significantly correlated with pCR were ADC GLCM Entropy (*p* < 0.001), DWI Volume (*p*:0.048), and ADC GLCM Entropy (*p*:0.045). From the multivariate logistic regression, the only parameter that remained significantly correlated with pCR in the training dataset was ADC GLCM Entropy (OR 0.14). The same analysis was repeated in the two validation datasets, confirming the significant correlation (respectively, OR 0.08 and OR 0.06) (see Appendix A and Figure 1). 

### 3.3. Calculation of Cut-Of

ROC curves were generated from the multivariate logistic regression analysis in both the datasets, and the AUC for pCR was 0.87 for the training dataset and, respectively, 0.92 and 0.88 for the two validation datasets (see Appendix A and Figure 2). The cut-off of ADC GLCM Entropy calculated on the training dataset was equal to −0.10. Using this cut-off, the sensibility of the model was 70%, the specificity of the model was 74%, and the accuracy of the model was 73%. The same cut-off, calculated on the two validation datasets, showed, retrospectively, a sensibility of 100% and 80%, a specificity of 66.7% and 80%, and an accuracy of 72.7% and 80% (see Figure 3 and Table 4). 

## 4. Discussion

The results of our study suggest that a non-invasive and dynamic analysis of MRI imaging, such as D-TA, can be used to predict the outcome of patients with LARC and, therefore, may help in the research on predictive biomarkers of responses to neoadjuvant C-RT and selection of patients who may avoid the risk of surgery. Many authors have already investigated the possible correlation of imaging techniques, especially MRI (volumetric analysis and TA) with patient outcomes; however, many of them mainly used T2 and DWI maps [23,31,76,77,78,79,80,81,82,83,84]. Conversely, CT imaging is used in different pathologies [85,86,87,88,89,90,91]. The results obtained in the present study show that TA applied to ADC maps could help to assess clinical and pathological response in LARC. In a previous study, significant findings in the response assessment were found about kurtosis, based on the T2-weighted dataset of images [92,93,94]. The authors found a significantly lower kurtosis in pre-treatment TA of pCR, with significant changes after CRT. Conversely, we chose to extrapolate TA parameters from ADC maps because of the direct relationship between the ADC values and tissue architecture, which seems to be independent of acquisition parameters. Indeed, some differences in geometrical and acquisition modality of the sequence could modify the spatial and contrast resolution of images, introducing a factor of variability. DWI sequence can also be influenced by the same parameters, but the obtained ADC maps are nonparametric dataset, similar to CT images, and their gray-scale levels in the image, are directly correlated to the imaged substrate (proton diffusivity and atomic number, respectively). At the same time, ADC sequences seem stable and useful in the setting of LARC [95,96]. We, therefore, found that the D-TA ADC GLCM Entropy was significantly decreased in patients who achieved pCR in the treatment. These results were in line with those recently reported by Nie et al., who found a higher Entropy (on T2 sequence) and a lower GLCM-Homogeneity (on DWI sequences) in LARC patients who did not benefit from neoadjuvant C-RT, in terms of a complete pathological response [97]. All together, these findings led us to hypothesize that the above-mentioned parameters might be strongly predictive of treatment resistance in LARC patients. Liu et al. [98] already used the ADC texture analysis for the same endpoint, suggesting that apparent diffusion coefficient texture analysis could improve the prediction of responders to neoadjuvant treatment. Similarly, other authors have successfully used ADC maps calculated on basal MRI for rectal cancer staging [99,100]. 

At present, MRI radiomic changes featuring between pre- and post-nC-RT are under active investigation in TRG prediction for pCR, good response, and T-downstaging [101]. For this purpose, Shayesteh et al. [64] assessed 96 delta radiomics features extracted from T2-weighted MRIs of 53 patients. They used 17 patients as an external validation set and reached, in the delta-radiomic-based model, the highest AUC of 0.96 (±0.01). Their delta-radiomics model outperformed both pre- and post-treatment features (*p*-value < 0.05).

In another model, Wan et al. [66] used a T2-weighted and DWI MRI showing a good performance for pCR prediction and reporting AUCs of 0.91 in the training and validation sets. Similarly, Jeon et al. [67] investigated both 2D and 3D features extracted from T2-weighted MRIs, finding that 2D delta-radiomics can be used as a good surrogate for 3D features in rectal cancer patients. 

In more recent times, Boldrini et al. [60] investigated the feasibility of radiomics analysis to predict cCR using a hybrid 0.35 T magnetic resonance, acquired during RT procedures. In this case, the variation in the three target delta features (energy, gray-level non-uniformity, and least axis length), showed a statistically significant correlation (*p*-value < 0.05) with pCR achievement. These data were later confirmed in the validation set by the same group, who reported a performance of ΔL least in predicting cCR and pCR, with an accuracy of 81% and 79%, respectively [62]. 

In line with this evidence, changes in tumor morphology and heterogeneity in delta radiomics features have also been correlated with the risk of metastatization and with the OS. In fact, Chiloiro et al. [63] selected 110 delta radiomics features showing the relevance of this approach in identifying the subset of patients with a higher risk of distant metastases at two years, in a large single-institution cohort. 

To our knowledge, our study represents the first attempt to predict pathological response in LARC patients using the delta radiomics approach based on the use of ADC sequences with volumetric TA, with three datasets. Furthermore, the use of delta radiomics features, with the calculation of the differences among features before and after specific treatment, would probably provide more detailed information about treatment response than static TA. The robustness of the delta radiomics signature was evaluated in a recent study on a phantom model, which confirmed an increased reproducibility of D-TA signature compared to TA extracted from ADV, T2w, and DWI MRI Scans [61]. Our results confirm, in a larger dataset, including three cohorts from different institutions, that the volumetric D-TA, and specifically, the ADC-MRI, could provide additional information in assessing pathological response and can be used to identify the patients at risk. 

### Limitations of the Study

The results of our study may be worthy of critical consideration for methodological and technical refinements due to the retrospective nature. We believe that the correlations between textural parameters and clinical outcomes may need further investigation to better understand the pathological basis sustaining the observed TA parameters. We definitely need to further investigate the real reproducibility and the reliability of this kind of analysis in other oncological centers, targeting different parameters of MRI acquisitions. We believe that the ability to perform subsequential imaging in each single center using the same acquisition parameters might increase the robustness of D-TA.

## 5. Conclusions

Pathological response to neoadjuvant CHT-RT represents an early outcome of treatment, strictly correlated with the prognosis of patients. Our results appear to be promising, since the D-TA seems to improve the knowledge of the predictive factors of response and may lead to different approaches to this subset of patients, such as intensive neoadjuvant chemotherapy or short-course RT with intensive adjuvant chemotherapy, or deferral of surgery, in selected subsets. Further prospective studies on a large population with external validation are needed to best estimate the present preliminary data. 

## Figures and Tables

**Figure 1 cancers-14-03004-f001:**
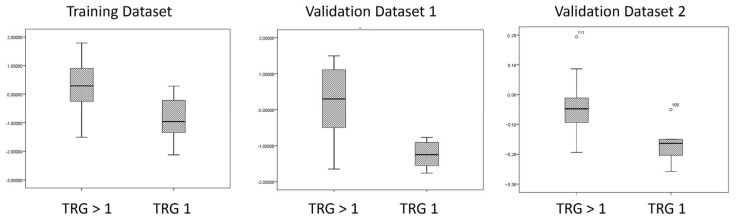
Box plot of the D-TA ADC GLCM Entropy in the training dataset (**left**) and validation datasets (**right**).

**Figure 2 cancers-14-03004-f002:**
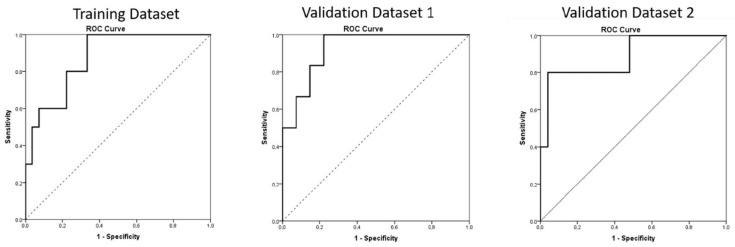
ROC Curve for the prediction of pCR (TRG 1) for training dataset (**left**) and validation datasets (**right**).

**Figure 3 cancers-14-03004-f003:**
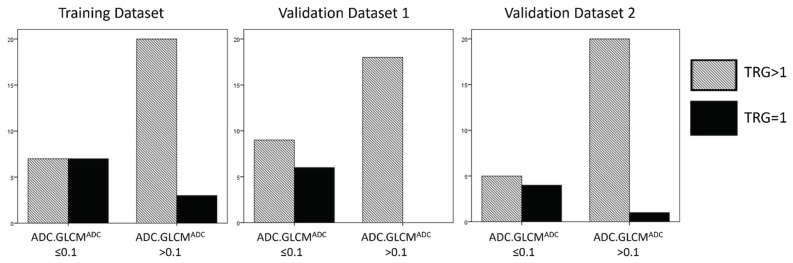
The cut-off of ADC GLCM Entropy calculated on the training dataset was equal to −0.10. Using this cut-off, the sensibility of the model in the training and in the two validation datasets, respectively, was 70%, 73%, and 100%, the specificity of the models was 74%, 66.7%, and 80%, and the accuracy of the model was 73%, 72.7%, and 80%.

**Table 1 cancers-14-03004-t001:** Acquisition parameters of the three magnetic resonance imaging vendors used in the three datasets.

Parameters	Training Dataset	Validation Dataset One	Validation Dataset Two
Vendor	Signa Excite HD, GE Healthcare 1.5 T	Signa Voyager HD, GE Healthcare 1.5 T	1.5T system, Achieva XR, Software release 5.3.1, Philips, Amsterdam, The Netherlands
Sequences	FSE T2 (axial, coronal, sagittal), T1 (axial pre and post c.e.), DWI and ADC	FSE T2 (axial, coronal, sagittal), T1 (axial pre and post c.e.), DWI and ADC	FSE T2 (axial, coronal, sagittal), T1 (axial pre and post c.e.), DWI and ADC
DWI	B 0. 500. 800 s/mm^2^	B 0. 500. 1000 s/mm^2^	B 0. 600. 1000 s/mm^2^

**Table 2 cancers-14-03004-t002:** Characteristics of patients.

Characteristic	Training Dataset	Validation Dataset 1	Validation Dataset 2	Chi-Square Test
**Sex**				*p*:0.597
Males	26 (70%)	21 (64%)	17 (57%)	
Females	11 (30%)	12 (36%)	13 (43%)	
**Age**				
<70 years	23 (62%)	22 (64%)	16 (53%)	*p*:0.841
>70 years	14 (38%)	11 (36%)	14 (47%)	
**Stage (T)**				*p*:0.340
cT2	8 (22%)	6 (18%)	3 (10%)	
cT3	25 (67%)	18 (55%)	20 (66%)	
cT4	4 (11%)	9 (27%)	7 (24%)	
**Stage (N)**				*p*:0.323
cN0	5 (14%)	4 (12%)	2 (7%)	
cN1/2	32 (86%)	29 (88%)	28 (93%)	
**Grading**				*p*:0.743
G1	2 (5%)	3 (9%)	2 (7%)	
G2	30 (81%)	23 (70%)	20 (67%)	
G3	5 (14%)	7 (21%)	8 (26%)	
**TRG**				*p*:0.180
1	10 (27%)	6 (18%)	5 (17%)	
2	13 (35%)	19 (57%)	13 (43%)	
3	13 (35%)	8 (24%)	8 (26%)	
4	1 (3%)	0 (0%)	4 (14%)	

**Table 3 cancers-14-03004-t003:** Univariate analysis (Chi-Square) of the reliable texture features and the chosen endpoint (TRG0) for the training dataset.

MRI Sequence	TA Parameter	Univariate Analysis (Chi-Square)	Bonferroni Correction (Number: 27)
T2-MRI	Volume.ml	0.49	NS
	Skewness	0.68	NS
	Sphericity	0.82	NS
	Compacity	0.21	NS
	GLCM.homogeneity	0.17	NS
	GLCM.entropy	0.97	NS
	GLCM.dissimilarity	0.62	NS
DWI-MRI	Volume.ml	0.00018	0.0486
	Skewness	0.03	NS
	Kurtosis	0.20	NS
	Entropy	0.25	NS
	Compacity	0.71	NS
	GLCM.homogeneity	0.45	NS
	GLCM.contrast	0.37	NS
	GLCM.correlation	0.72	NS
	GLCM.entropy	0.0017	0.0459
	GLCM.dissimilarity	0.32	NS
ADC-MRI	Volume.ml	0.54	NS
	Skewness	0.98	NS
	Kurtosis	0.90	NS
	Entropy	0.42	NS
	Energy	0.59	NS
	Sphericity	0.78	NS
	Compacity	0.11	NS
	GLCM.homogeneity	0.03	NS
	GLCM.contrast	0.40	NS
	GLCM.entropy	0.00017	0.00459
	GLCM.dissimilarity	0.60	NS
Clinical Parameters	Sex	0.32	NS
	Age	0.25	NS
	Stage	0.24	NS
	Grading	0.28	NS

**Table 4 cancers-14-03004-t004:** Specificity, sensibility, and accuracy of the cut-off of the texture parameter GLCM. Entropy^ADC^.

Model	Count	TRG > 0	TRG0	Total
Training Dataset	GLCM.Entropy^ADC^ < 0.10	7	7	14
	GLCM.Entropy^ADC^ > 0.10	20	3	23
Validation Dataset 1	GLCM.Entropy^ADC^ < 0.10	9	6	15
	GLCM.Entropy^ADC^ > 0.10	18	0	18
Validation Dataset 2	GLCM.Entropy^ADC^ < 0.10	5	4	9
	GLCM.Entropy^ADC^ > 0.10	20	1	21
Legend	GLCM.Entropy^ADC^ < 0.10	False Positive	True Positive	
	GLCM.Entropy^ADC^ > 0.10	True Negative	False Negative	

## Data Availability

The data presented in this study are available on request from the corresponding author. The data are not publicly available due to privacy restrictions.

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
