# Peer review of "Ability of Delta Radiomics to Predict a Complete Pathological Response in Patients with Loco-Regional Rectal Cancer Addressed to Neoadjuvant Chemo-Radiation and Surgery"

_cancers, 2022, doi:10.3390/cancers14123004_

Round 1
Reviewer 1 Report
Good article with sound analysis of obtained results. I have no principal critisim. Please check wording and misprints, e.g. line 28.... petodological items, line 101 retroospective study, line 119 all of th epatinets
Author Response
Good article with sound analysis of obtained results. I have no principal critisim. Please check wording and misprints, e.g. line 28.... petodological items, line 101 retroospective study, line 119 all of th epatinets
- We thank the Reviewer for his/her time spent reading and evaluating our manuscript. We have corrected many typos and misprints.
Reviewer 2 Report
I think the work is very interesting. Longitudinal analysis of radiomic data can be really useful for predicting the response to treatments. In particular, these results are of great importance in the neoadjuvant setting.
Here are my observations.
Firstly, there are a lot of typos and English could be improved.
I understand that, although there are 3 different centers, all patients were analyzed by the same MRI. This limits the validity of these results, infact the two datasets are not real test sets, but internal validation sets. If, on the other hand, the RM machines are different, it must be specified better.
in fact, such a high accuracy in all datasets is impressive, if this is not affected by the use of other magnetic resonance machines. Furthermore, this result looks even more impressive with a training set of just 37 patients.
If, on the other hand, it is correct to deduce that all the resonances were performed by the same machine, then it makes little sense to divide the cohort into 3 datasets, as none can function as a test set, but only as an internal validation set, in this case a single validation set is more than enough.
Furthermore, by combining the different datasets, it is possible to start thinking about the application of artificial intelligence algorithms.
in conclusion, I think the work is really very interesting, it responds to an unmeet clinical need. But I think this is only the first step of the research. Having an external dataset would greatly increase the relevance of this paper.
If the MRI machines are 3 different, then, this work has great relevance, but it needs to be explained better, because in this version it is not very clear.
Author Response
I think the work is very interesting. Longitudinal analysis of radiomic data can be really useful for predicting the response to treatments. In particular, these results are of great importance in the neoadjuvant setting.
- We thank the Reviewer for his/her time spent reading and evaluating our manuscript.
Here are my observations.
Firstly, there are a lot of typos and English could be improved.
- We have corrected many typos in our manuscript and we apologize for not checking it before first submission.
I understand that, although there are 3 different centers, all patients were analyzed by the same MRI. This limits the validity of these results, infact the two datasets are not real test sets, but internal validation sets. If, on the other hand, the RM machines are different, it must be specified better.
In fact, such a high accuracy in all datasets is impressive, if this is not affected by the use of other magnetic resonance machines. Furthermore, this result looks even more impressive with a training set of just 37 patients.
If, on the other hand, it is correct to deduce that all the resonances were performed by the same machine, then it makes little sense to divide the cohort into 3 datasets, as none can function as a test set, but only as an internal validation set, in this case a single validation set is more than enough.
Furthermore, by combining the different datasets, it is possible to start thinking about the application of artificial intelligence algorithms.
- We thank the Reviewer for this interesting point. Actually the patients were analysed with 3 different MRI machines. We have better specified it in the Methods section. In our experience, delta-radiomics features are more reliable in multicentre analysis than classic radiomics features. We have also performed an phantom study to demonstrate this (Nardone et al. Delta-radiomics increases multicentre reproducibility: a phantom study. Med. Oncol. 2020). This preclinical experiment could explain the results obtained with the small cohort analysed. We agree also with the potential application of AI algorithms, that are going to be the next step of our research.
In conclusion, I think the work is really very interesting, it responds to an unmeet clinical need. But I think this is only the first step of the research. Having an external dataset would greatly increase the relevance of this paper.
If the MRI machines are 3 different, then, this work has great relevance, but it needs to be explained better, because in this version it is not very clear.
- We thank the Reviewer. As above, we have better specified the use of different machines for the three datasets analysed.